Geographic source of bats killed at wind-energy facilities in the eastern United States

Wieringa Jamin G. jamingwieringa@gmail.com 1 2
Nagel Juliet 3
Campbell C.J. 4 5
Nelson David M. 3
Carstens Bryan C. 1
Gibbs H. Lisle 1 2
1 Department of Evolution, Ecology and Organismal Biology, Ohio State University , Columbus , OH , United States of America
2 Ohio Biodiversity Conservation Partnership , Columbus , OH , United States of America
3 Appalachian Lab, University of Maryland - Center for Environmental Science , Frostburg , MD , United States of America
4 Department of Biology, University of Florida , Gainesville , FL , United States of America
5 Bat Conservation International , Austin , TX , United States of America
Sunny Armando
Electronic publication date: 2024 Feb 5
Publication date: 2024
Volume: 12
Electronic Location ID: e16796
Received 2023 Oct 23; Accepted 2023 Dec 24
Copyright: ©2024 Wieringa et al.
Copyright year: 2024
Copyright holder: Wieringa et al.
License: This is an open access article distributed under the terms of the Creative Commons Attribution License, which permits unrestricted use, distribution, reproduction and adaptation in any medium and for any purpose provided that it is properly attributed. For attribution, the original author(s), title, publication source (PeerJ) and either DOI or URL of the article must be cited.
License URL: https://creativecommons.org/licenses/by/4.0/

Keywords: Wind-energy facilities, Bats, Conservation, Isotope, Migration

Funding: The Competitive State Wildlife Grants Program to Ohio State University GRT00046616 The University of Maryland Center for Environmental Science as jointly administered by the US Fish and Wildlife Service, the Ohio Division of Wildlife and the Maryland Department of Natural Resource This work was supported by a grant (GRT00046616) from the Competitive State Wildlife Grants Program to Ohio State University and the University of Maryland Center for Environmental Science as jointly administered by the US Fish and Wildlife Service, the Ohio Division of Wildlife and the Maryland Department of Natural Resources. The funders had no role in study design, data collection and analysis, decision to publish, or preparation of the manuscript.

==============================
Bats subject to high rates of fatalities at wind-energy facilities are of high conservation concern due to the long-term, cumulative effects they have, but the impact on broader bat populations can be difficult to assess. One reason is the poor understanding of the geographic source of individual fatalities and whether they constitute migrants or more local individuals. Here, we used stable hydrogen isotopes, trace elements and species distribution models to determine the most likely summer geographic origins of three different bat species (Lasiurus borealis, L. cinereus, and Lasionycteris noctivagans) killed at wind-energy facilities in Ohio and Maryland in the eastern United States. In Ohio, 41.6%, 21.3%, 2.2% of all individuals of L. borealis, L. cinereus, and L. noctivagans, respectively, had evidence of movement. In contrast, in Maryland 77.3%, 37.1%, and 27.3% of these same species were classified as migrants. Our results suggest bats killed at a given wind facility are likely derived from migratory as well as resident populations. Finally, there is variation in the proportion of migrants killed between seasons for some species and evidence of philopatry to summer roosts. Overall, these results indicate that the impact of wind-energy facilities on bat populations occurs across a large geographic extent, with the proportion of migrants impacted likely to vary across species and sites. Similar studies should be conducted across a broader geographic scale to understand the impacts on bat populations from wind-energy facilities.

Introduction

In recent decades, renewable-energy production has increased worldwide, helping to reduce greenhouse gas emissions and mitigate global climate change. However, renewables have also had unexpected negative environmental impacts, such as wildlife fatalities (Katzner et al., 2019). For example, estimates based on the buildout of wind-energy facilities in North America from a decade or more ago, suggest that hundreds of thousands of bats are killed annually at these sites (Arnett et al., 2008; Arnett & Baerwald, 2013; Hayes, 2013). Most of these fatalities are from three species of tree roosting, migratory bats (silver-haired bats, Lasionycteris noctivagans; hoary bats, Lasiurus cinereus; and eastern red bats Lasiurus borealis; hereafter termed “tree-roosting bats” sensu Griffin, 1970), and levels of mortality may impact the viability of these species during the coming decades (Frick et al., 2017; EPRI Electric Power Research Institute & Friedenberg, 2020; Friedenberg & Frick, 2021).

Most tree-roosting bat fatalities at wind-energy facilities occur during the late summer and autumn migration period (Kunz et al., 2007; Arnett et al., 2008; Taber & Butryn, 2018). Similarly, other migratory bat species also experience high proportions of fatalities at wind-energy facilities elsewhere in the world (Barros, Magalhaes & Rui, 2015; Rydell et al., 2010; Voigt et al., 2012). Thus, migration is thought to play an important role in the interactions of bats with wind-energy facilities. However, the geographic extent across which wind-energy facilities impact migratory bats remains poorly characterized (Voigt et al., 2012). This lack of understanding makes it challenging to mitigate the effects of these fatalities. For example, current fatality estimates do not typically distinguish between whether migratory or resident bat populations are affected, where migratory individuals originate, or how these factors vary through space and time. Such information could help to identify the species and populations at most risk from current or planned wind-energy development and thus help inform conservation and management decisions.

Stable hydrogen isotope values (δ2H) in bat fur are intrinsic biomarkers markers that can be used to help identify the summering grounds of tree-roosting bats killed at wind-energy facilities (Baerwald, Patterson & Barclay, 2014; Pylant et al., 2016). Such data suggest the proportions of individuals that summered in distinguishably distant regions (hereafter, “migrants”) versus those that did not (hereafter, “residents”) may vary across the species and sites studied (Baerwald, Patterson & Barclay, 2014; Cryan Stricker & Wunder, 2014; Pylant et al., 2016; Fraser, Brooks & Longstaffe, 2017). To clarify, when we use the term “resident,” we apply it given an absence of evidence that a bat traveled a long distance. We used this language as we are unable to distinguish between true residents and those that might be migrants but show similar chemical signatures as residents. Our classification of migratory versus resident is of necessity course due to the limited resolution of our method of sourcing individuals (see below). Despite this limitation, this information is valuable because prior studies have been conducted on just one or two of the three affected species of tree-roosting bats, using samples from different sites, years, and seasons. This has made it difficult to assess intra- and inter-specific variation in geographic origins at individual wind energy sites. Furthermore, a known limitation of using δ2H values is that precipitation can vary primarily along latitudinal and elevational gradients (Bowen, Wassenaar & Hobson, 2005). Therefore, an individual classified based on δ2H as being a resident without evidence of movement could have summered at a site longitudinally distant from the location where it was killed. This means that δ2H-based analyses likely underestimate the proportions of individuals that are mid- to long-distance migrants. New approaches, such as combining δ2H and trace element data with predictions of species distributions, promise to help improve the accuracy and precision of geographic assignments of tree-roosting bats killed at wind-energy facilities in North America (Wieringa et al., 2023; Kruszynski et al., 2021).

Here we use a new approach to source bats that combines δ2H, trace element, and species distribution models (Araújo & Guisan, 2006) to determine the geographic origin of tree-roosting bats killed at utility-scale wind-energy facilities in the midwestern (Ohio) and Appalachian Mountain (Maryland) regions of the United States. These areas were selected because (1) they represent two distinct geographic areas, (2) wind-energy facilities in these states have relatively high rates of tree-roosting bat fatalities and (3) we had access to fur samples from bat carcasses from these states. The goals of our study were to: (1) determine the migratory status of bats killed; (2) investigate the how the proportion of migratory individuals killed varies within and across years; and (3) infer the most likely origin of migrants and the distance and direction they most likely traveled. In Maryland, we also compared our results based on multiple marker analyses to a previous study that used only δ2H values (Pylant et al., 2016; Campbell et al., 2020). These data allowed us to address the hypotheses that most bats killed are migratory, that migratory proportions are consistent among species, sites, years, and seasons, and that migrants originate north of their location of death.

Materials and Methods

To determine the likely origin of tree-roosting bats killed at wind-energy facilities in Ohio and Maryland, we combined information from assignment surfaces generated from trace elements, δ2H, and species distribution models (Wieringa et al., 2023). We first created species distribution models (SDMs) for summer months (June and July; Wieringa, Carstens & Gibbs, 2021) which is when the fur of these species is likely grown (Cryan et al., 2004; Fraser, Longstaffe & Fenton, 2013; Pylant, Nelson & Keller, 2014; Fraser, Brooks & Longstaffe, 2017). Similarly, we collected chemical markers from the fur of bats, as outlined below. These were then combined to determine the geographic origin of bats killed at wind-energy facilities.

Sample collection

Fur samples were taken from carcasses of silver-haired, hoary, and eastern red bats collected during post-construction monitoring activities at two wind-energy facilities in northwestern Ohio and one in western Maryland (Table 1). Samples were collected in 2012 and 2017 in Ohio and in 2014, 2015, 2016, and 2018 in Maryland. In Ohio, samples were taken from three seasons (Spring, Summer, and Fall) and in Maryland during two seasons (Spring and Fall). Note that these numbers reflect a subset of the total from each wind-energy facilities and have not been corrected for search bias and cannot be used to assess fatality rates or other related metrics.

Table 1 Sample breakdown my season, year, and location.

Species	Location	Season	Year	# of samples	
L. borealis	Ohio	Spring	2012	14	
2017	2	
Summer	2012	43	
2017	17	
Fall	2012	0	
2017	13	
Maryland	Spring	2016	3	
2018	2	
Fall	2016	37	
2018	2	
L. cinereus	Ohio	Spring	2012	14	
2017	7	
Summer	2012	6	
2017	13	
Fall	2012	1	
2017	6	
Maryland	Spring	2014	0	
2016	5	
2018	1	
Fall	2014	3	
2016	18	
2018	0	
L. noctivagans	Ohio	Spring	2012	3	
2017	17	
Summer	2012	0	
2017	4	
Fall	2012	7	
2017	14	
Maryland	Spring	2011	1	
2015	1	
2016	2	
Fall	2011	26	
2016	12	
2018	2	

Hydrogen isotope and trace element determination

Fur samples were cleaned as described in Pylant et al. (2016) using 1:200 Triton X-100 detergent, 100% ethanol, and then air-dried. The δ2H value of non-exchangable hydrogen in each sample was analyzed using a comparative equilibration approach (Wassenaar & Hobson, 2003). For this approach fur samples, international standards (USGS42, USGS43, CBS, KHS) and an internal keratin standard (porcine hair and skin, Spectrum Chemical product # K3030, New Brunswick, NJ, USA) were weighed into silver capsules and exposed to ambient for >72 h to permit equilibration of exchangeable hydrogen in keratin. The capsules were then folded and loaded into a Costech Zero-Blank autosampler (Valencia, CA, USA), which was repeatedly purged with dry helium.

Samples were analyzed for δ2H values using a Thermo Fisher high temperature conversion/elemental analyzer (TC/EA) pyrolysis unit interfaced with a ThermoFisher Delta V+ isotope ratio mass spectrometer (Thermo Scientific, Bremen, Germany) via a Thermo Fisher Scientific ConFlo IV universal continuous flow interface. Measured values of δ2H were normalized to the Vienna Standard Mean Ocean Water-Standard Light Antarctic Precipitation scale using USGS42, USGS43, CBS (caribou hoof standard), and KHS (kudu horn standard) standards. The δ2H values of non-exchangeable hydrogen of these standards are −72.9, −44.4, −157.0, and −35.3‰, respectively (Coplen & Qi, 2016; Soto et al., 2017). The long-term δ2H value of the internal keratin standard at CASIF is −59.5 ± 2.3‰.

For trace element analysis, protocols from Wieringa et al. (2020) were followed. In short, fur samples were cleaned and digested in acid with a gold trichlorite stabilizer and diluted to 10ml after cooling to room temperature. A 10ppb indium internal standard was included in all samples. Samples were then placed on a Thermo Finnigan Element 2 High Resolution ICP-MS at the Trace Element Research Lab (TERL) at The Ohio State University. We then corrected each sample using the internal standard using the integrated software.

Assignment of geographic origin

Assignment of origin was determined as in Wieringa et al. (2023). Briefly, likely origin from three approaches (trace elements, stable hydrogen isotopes, and species distribution models) were combined using a cross validated combined canonical (CVCC) approach (Rundel et al., 2013) creating a single likely origin map. Each species was evaluated independently. For L. borealis and L. cinereus, this combination included all three probability layers of trace elements, stable hydrogen isotope values, and SDMs raised to the powers of 0.2, 0.6, and 1, respectively. In contrast, for L. noctivagans the best combination only used isotopes and SDMs, each raised to the power of 0.2 and 0.6, respectively. Details for this process and the creation of each layer can be found in Wieringa et al. (2023), Wieringa, Carstens & Gibbs (2021), Wieringa et al. (2020), Campbell et al. (2020), and Pylant et al. (2016). The resulting surface for each individual layer was normalized to sum to 1.

Next, we used the ‘oddsAtSamplingLocation’ function in the R package ‘isocat’ (Campbell et al., 2020) to determine the probability that the individual originated from the same area it was collected from. As the accuracy and prediction area can vary based on the thresholds set, we used the thresholds determined by Wieringa et al. (2023) that gave an expected accuracy of 80%. Setting this expectation for accuracy we can determine thresholds at which that accuracy is obtained when using known origin individuals, as was done in Wieringa et al. (2023). When we use the validated thresholds to obtain an accuracy of 80% for L. borealis, L. cinereus, and L. noctivagans these probability thresholds were 0.265, 0.0000327, and 0.1713, respectively. This procedure thus provides an estimate of probability of local origin that incorporates the uncertainty associated with estimating the geographic provenance for turbine-killed samples of unknown origin. If an individual is determined to be below the threshold, then it was classified as migratory relative to the wind-energy facility as their probability of originating near the wind farm was likely below 80%, whereas individuals above the threshold were classified as resident. For individuals classified as “resident”, as defined by the approach above, we cannot exclude the possibility that they summered near the site of their death. Due to the volant nature of these bats we also added adjustments to account for local movements by individuals. For example, Weller et al. (2016) recorded a single hoary bat flying 68 km in a single night. Others have stated nightly flights of ∼10 km (Hayes & Wiles, 2013). As a result, we took the mean of those two values (39 km), and if an area above the threshold was within 39 km it was reclassified as a resident individual. To determine if there are differences in seasonal proportions of migrant vs. resident individuals we used ‘prop.test’ (R Core Team, 2013)

Source location and direction

Following Cryan Stricker & Wunder (2014), we also investigated the direction and distance of the most likely origin using only migratory bats. Some previous work used known capture locations and the highest predicted origin cell to determine likely migration movements for L. cinereus (Cryan Stricker & Wunder, 2014) and we implemented this approach as a result. However, this approach can result in the highest value cell and known origin being hundreds of kilometers apart (Campbell et al., 2020). To remedy this we also averaged the location of the most likely origin cells in the 95th percentile and above (Wieringa et al., 2023) to hopefully get a better representation of the likely origin. Using the averaged location of 95th percentile and above cells potentially gives a better estimate of the origin due to more data included. This averaged location was then used to determine the direction (azimuth) between the sampling location and averaged origin we used the ‘bearing’ function in the R package ‘geosphere’ (Hijmans et al., 2017).

Results

Ohio

In Ohio, for L. borealis, L. cinereus, and L. noctivagan s, respectively, we observed that 41.6%, 21.3%, and 2.2% of all individuals killed lacked evidence of movement, and thus were classified as migrants (Table 2). To be clear, the term migrants as used here are those that had clear signatures of non-local origin. Across three migratory periods (spring, summer, and fall) in Ohio, there were no differences in the proportion of migrant bats killed between seasons for L. borealis (prop.test in base R; p = 0.75) or L. noctivagans (p = 0.56), but there were differences for L. cinereus, with spring migration having 23.8% migrants, summer having 15.8% migrants, and fall having 57.1% migrants (p = 0.02). No species exhibited differences in the proportion of bats determined to be migrants between years sampled (Fig. 1).

Table 2 Summary of results for each species.

Columns 3–6 show the percent of samples from that migratory time period that are determined to be local. A higher percent means more individuals were determined to be resident vs migratory. Columns 7–10 show the distance and direction metrics determined using the highest value cell and averaged 95th percentile cell.

Region	Species	Percent local	Spring local	Summer local	Fall local	Average distance
to 95th percentile	Average direction
to 95th percentile	Average distance
to max cell	Average direction
to max cell	
Ohio	L. borealis	0.5843	0.5	0.6	0.6154	1560.78 (637.71)	45.16 (20.63)	1184.91 (742.03)	329.56 (78.38)	
L. cinereus	0.7872	0.7619	0.8421	0.4286	2143.38 (139.73)	82.47 (4.67)	2063.44 (373.37)	79.04 (6.99)	
L. noctivagans	0.9778	1	1	0.9474	2368.598 (0)	64.44 (0)	2182.57 (0)	72.59 (0)	
Maryland	L. borealis	0.2273	0.4	NA	0.2051	1318.32 (82.61)	46.04 (1.19)	1183.33 (236.96)	30.22 (9.86)	
L. cinereus	0.6296	0	NA	0.8095	2712.61 (79.24)	84.05 (3.95)	2538.18 (418.84)	81.33 (8.09)	
L. noctivagans	0.7273	1	NA	0.7	2906.01 (33.48)	79.78 (6.02)	2622.38 (87.70)	79.76 (2.58)	

Figure 1 Results for Ohio.

Pie charts showing the changes in proportions of resident (orange) vs. migratory (blue) wind-energy facilities mortality in Ohio, including overall mortality, seasonal differences, and yearly changes. In addition, we show the direction of travel from the highest value cell (red) or average 95th percentile (blue) location to the known sampling location for all individuals. Significant differences are highlighted with a black *.

In terms of the direction of travel by non-local, migratory individuals, for L. borealis the highest value cell to the wind-energy facilities had an average bearing of 30.22° (SD: 9.85°; 0/360° is due north, 180° due south), whereas the average direction of 45.16° (SD: 1.92°) from the average 95th percentile cell to the wind-energy facilities. For L. cinereus we found an average bearing of 79.04° (SD: 6.99°) to highest value cell and to the average 95th percentile cell an average bearing of 82.47° (SD: 4.67°). Finally, the one migratory L. noctivagans identified in Ohio had a bearing of 72.59°, whereas the average 95th percentile cell was 64.44° (Fig. 1).

Maryland

In Maryland, for L. borealis, L. cinereus, and L. noctivagans, respectively, we observed 77.3%, 37.1%, and 27.3% of the mortalities have evidence of movement and thus were considered migrants (Fig. 2). Between the spring and fall migration periods there were not differences in the proportion of migrant bats for L. borealis (p = 0.68) or L. noctivagans (p = 0.49), but there were relatively less migrant L. cinereus found in spring (0.0%) than to fall (81%; p = 0.001). Like in Ohio, we did not observe changes in the proportion of bats determined to be migratory between years sampled.

Figure 2 Results for Maryland.

Pie charts showing the changes in proportions of resident (orange) vs migratory (blue) wind-energy facilities mortality in Maryland, including overall mortality, seasonal differences, and yearly changes. In addition, we show the direction of travel from the highest value cell (red) or average 95th percentile (blue) location to the known sampling location for all individuals. Significant differences are highlighted with a black asterisk (*).

In terms of the direction of travel by non-local, migratory individuals, for L. borealis the highest value cell to the wind-energy facilities had an average bearing of 30.22° (SD: 9.86°), whereas the average direction of 46.04° (SD: 1.92°) from the average 95th percentile cell to the wind-energy facilities. For L. cinereus, the highest value cell to the wind-energy facilities had an average direction of 81.34° (SD: 5.08°) and the average direction of 84.05° (SD: 3.95°) for the average 95th percentile cell to the wind-energy facilities. Finally, for L. noctivagans the direction to the highest value cell was 79.76° (SD: 2.58°), whereas the average 95th percentile cell was in the average direction of 79.78° (SD: 6.02°) (Fig. 2).

Discussion

In this study we used two biomarkers and species distribution models to determine the migratory status of tree-roosting bats killed at wind-energy facilities in Ohio and western Maryland. We stress that “migrant” as used here does not conclusively exclude the possibility that any of these individuals are not resident, but rather that we lack evidence for local origin using these methods. Further, we cannot exclude the possibility that some deemed “resident” were migratory. This could be due to similar chemical signatures for true residents and those that we cannot distinguish from true residents but could be migratory. Except for L. borealis in Maryland, we observed a trend in which most individuals of all species were not associated with evidence of movement, hereafter, likely “residents”. Furthermore, in both states the lowest proportion of migrants were individuals of L. noctivagans and the highest proportion of migrants were individuals of L. borealis, and the Maryland site had consistently had more migratory mortalities. Only for L. cinereus was there variation in the proportion of migrants across seasons.

One area that is important to acknowledge is the spatial uncertainty that is inherent to this approach (Wieringa et al., 2023). In short, we used thresholds that maintained an 80% accuracy. While we could have used thresholds that reduced the predicted area, we would have lost accuracy as a result. We felt it was more important to be correct in our prediction than to predict less area. In addition, there is some uncertainty due in the exact location of the most likely origin. This is due to us utilizing SDM’s in this approach. SDM’s are species wide and will skew results to the most suitable areas for the entire species, while this study is focused on the behavior of individuals. While this is true, incorporating SDM’s into the approach reduced predicted origin area while maintaining accuracy. This limited some of the inferences we could make as the highest probability and top 5% probability cells would be skewed to those areas for every individual. In the end, we chose to reduce our prediction area to increase our confidence in the question at hand.

It is important to acknowledge that our definition of what constitutes the resident geographic origin of an individual bat dependents on the resolution of the biomarkers used in our analyses. This uncertainty limits the spatial scale of that we can practically determine the impact of a given facility. Nonetheless, combining approaches represents a step forward in refining the geographic catchment of the impacts of wind-energy facilities on populations of the most impacted species in the United States as we can predict less area (Wieringa et al., 2023). Further analyses using additional biomarkers, such as strontium isotopes, could lead to further refinement of the scale of these impacts (Brewer, Rauch-Davis & Fraser, 2021).

Previous studies investigating the migratory status of L. borealis killed at wind-energy facilities have found that there is often variation between sites in the proportion of individuals that are determined to be resident vs. migratory. For example, 71% and 57% of dead L. borealis at wind facilities were determined to be of migrants from outside Illinois (Murtaugh et al., 2019) and Appalachia (Pylant et al., 2016), respectively. We found a lower proportion of individuals to be determined migrants at 58.4% in Ohio relative to Maryland (22.7%). Such differences in proportions of migrants may result from differences in the number of migrating bats passing through each region. Wieringa, Carstens & Gibbs (2021) hypothesized that there would be migratory pathways near the slopes of the Appalachian Mountains and near rivers. Study sites for the previous studies (Pylant et al., 2016; Murtaugh et al., 2019) fall into each of these proposed migratory pathways, whereas Ohio was shown to potentially be at the northern end of the migratory movements suggesting that migratory bats may be less common relative to resident individuals (Wieringa, Carstens & Gibbs, 2021).

Differences across regions in geographic origin also exist for L. cinereus. Baerwald, Patterson & Barclay (2014) suggested most of the individuals of L. cinereus killed at wind-energy facilities were migrant in Southern Alberta, whereas Pylant et al. (2016) showed that only ∼1% of L. cinereus killed at wind-energy facilities were migrants in Maryland. We also observed some differences between our two study sites. In Ohio, we observed an overall migrant proportion of 21.3%, while in Maryland we observe a 37.1% proportion of resident individuals. The difference between the two regions is clearest in the fall migration period, which could be due to different migration patterns in each area. The overall values of our study compared with Pylant et al. (2016) for Appalachia are similar, especially considering we are using additional markers to refine the area of assignment (Wieringa et al., 2023). The fact that most of mortalities collected the fall for L. cinereus are migratory could be due to possible movements from Canada into Ohio as was modeled in Wieringa, Carstens & Gibbs (2021), meaning that individuals could be migrating south through this region. This pattern was also seen in Wieringa, Carstens & Gibbs (2021) with L. cinereus moving from northern regions through Northwest Ohio.

Finally, for L. noctivagans we found results that differ from other regions (Baerwald, Patterson & Barclay, 2014). Overall, we found that 2.2% and 27.3% of bats killed by wind-energy facilities in Ohio and Appalachia, respectively, were found to be of migratory origin. In Baerwald, Patterson & Barclay (2014) they found that the majority of bats killed at wind farms were migratory. While L. noctivagans is migratory (Fraser, Brooks & Longstaffe, 2017), it is also known to overwinter at higher latitudes (Izor, 1979) and may be better classified as a partially migratory species (Wieringa, Carstens & Gibbs, 2021). Our results support this interpretation: while we observe some migrants, most individuals are still classified as resident.

One of the goals for this project was to have clear comparison between the utilization of the approach utilized here and the singular isotope approach that has been implemented previously. One way for us to address this is a comparison between this study and Pylant et al. (2016). We can utilize this study for comparison as we used the same site (Maryland) and some of the same samples. When comparing the results, we can see differences that result from the refinement of isotopic determination of likely origin using trace elements and distribution mapping techniques (Wieringa et al., 2023). For L. borealis, we found 77.3% to be of migratory while Pylant et al. (2016) found 57% to be of that status. Similar trends can be found for L. cinereus in that here we found 37.1% to be migratory while Pylant et al. (2016) found ∼1% to be the same. We clearly observed the same trend in that, in Maryland, of the two Lasiurus species more L. borealis killed at wind farms are migratory than L. cinereus. The specific differences in those numbers are likely due to the improved ability to determine origin, as was found in Wieringa et al. (2023).

We also investigated evidence for migration philopatry. Philopatry is defined when individual bats return to the same location year after year (Kurta & Murray, 2002). The fur that we sampled was grown during the summer (Fraser, Longstaffe & Fenton, 2013) and would therefore reflect the chemical makeup of the previous year’s summering grounds when an individual is sampled during the spring migration. For both locations sampled we found 100% of L. noctivagans individuals sampled in the spring to be residents. There are two possible explanations for this pattern: either the individuals overwintered in the indistinguishably similar summering grounds or individuals returned to the same summer grounds. We cannot discriminate between these two explanations with our existing data. However, due to the high proportion of fall individuals that were resident and since it is known that L. noctivagans can hibernate at high latitudes (Kurta et al., 2018), we suggest it is more likely these individuals overwintered in these areas.

The other two species showed more moderate levels of resident individuals during spring migration periods. For L. borealis, we found that the proportion of migrants was 50.0% and 60.0% in Ohio and Maryland. While figures on estimated philopatry are, to the best of our knowledge, not available for L. borealis other species of bats have shown similar rates of philopatry between years (Kurta & Murray, 2002). Some of these individuals could be due to overwintering in the local regions, as some distribution studies (Cryan, 2003) have had samples from these areas during winter months. Whitaker Jr, Rose & Padgett (1997) also suggested L. borealis can overwinter in regions where freezing temperatures are frequently encountered. As for L. cinereus, in Ohio we found a much lower rate of migrants (23.8%) while finding no resident individuals in Maryland during spring. Again, while this could be due to some level of philopatry and/or overwintering in Ohio, we are unable to tease these apart. Likely, it is some combination of the two, as Wieringa, Carstens & Gibbs (2021) did show that regions of Ohio could be suitable habitat during the winter months.

Regarding migration direction, our results suggest movements that differ from our hypothesis. Previous studies have found clear evidence of north-to-south movement (e.g., (Baerwald, Patterson & Barclay, 2014) but we did not observe this north-to-south pattern for our migratory individuals. This lack of a north-to-south pattern is likely an artifact of the approach as the highest value cell may not, in fact, be the best representation of the most likely origin (Campbell et al., 2020). This is especially true when we include SDMs in our analyses as they will predict species-wide relative habitat suitability for all individuals, rather than the most likely origin of a given individual. As a result, we are likely skewing our directions and distances due to this artifact in our analysis. The reason this approach can skew some aspects of the results but not affect overall accuracy are due to the limitations inherent to this approach. We determined our thresholds to obtain an accurate determination of origin above that threshold, not based on the origin being one of the highest probability cells. Thus, when we use the highest probability cells (or an average of the top 5% of cells) we may excluding the true origin. When SDM’s are included, this will inherently occur as the SDM is species wide and determining the likely origin of the entire species, while we are attempting to determine the origin of an individual. We choose to still include the SDM as it increased the overall performance for overall accuracy which we deemed the most important. While this may affect our results, combining these approaches does provide a better overall precision at the same accuracy level (Wieringa et al., 2023), which may prove an acceptable tradeoff in model precision and accuracy (Campbell et al., 2020).

Our observation that bats killed at wind-energy facilities can be from both migratory and resident populations has significant conservation implications. Of greatest importance, our results suggest that wind-energy facilities have significant impacts on local bat populations for species most impacted by wind facilities across seasons. This implies several things for the conservation of these species. For example, it may indicate that other factors aside from migratory status (Thaxter et al., 2017) play a role in bat mortality (Roemer et al., 2017). Regardless of other factors, still the best mitigation approach is curtailment. A common approach to reduce wind-energy facilities mortality of bats is to altering cut-in speed during migration time periods (Baerwald et al., 2009; Martin et al., 2017). While most bat mortality is thought to occur during presumed migration time frames (Arnett et al., 2008), our results suggest that impacts from wind-energy facilities could occur over longer time periods. Some studies have found, that for some species, there is a steady rate of collision mortality in migratory bats (Piorkowski & O’Connell, 2010). Due to this result, these types of approaches need to be applied over the entire year to conserve resident populations as the impacts of wind-energy facilities may not be limited to just presumed migration time periods and these results could be used to optimize the conservation impacts either through modulating blanket cut-in speed (Adams, Gulka & Williams, 2021; Hayes et al., 2019) or using more complex algorithmic approaches (Barré et al., 2023).

Finally, our findings that wind-energy facilities impact on both migratory and resident individuals has important implications for the management of these species. We show that wind-energy facilities impact individuals with diverse geographic origins, highlighting that bat fatalities at wind-energy facilities likely have far-reaching geographic impacts on populations. We also found fatalities of resident individuals outside of the autumn migration period, indicating fatalities at wind-energy facilities may impact populations during their spring migration and summer maternity seasons. Further investigations are needed to determine the magnitude and extent of impacts wind energy facilities have on bat populations. Further, we found that patterns of bat migration vary both spatially and temporally. We observed differences between two locations (Ohio and Maryland) that highlight likely differences in migratory route and structure between the two regions. Our results highlight the urgent need for improved understanding of the relationships between migration ecology and population-level consequences of wind-energy associated fatalities in these species.

We thank the Trace Element Research Lab (TERL) at The Ohio State University for their assistance with the processing of samples. Specifically, we thank Anthony Lutton for help with scheduling lab equipment use and for help with interpreting our results and John Olesik and other members of the TERL for general assistance. We also thank the Carstens and Gibbs labs in the Department of EEOB at Ohio State for their advice and assistance with the editing of this manuscript. Finally, we thank Erin Hazelton and Jonathan Sorg, Ohio Division of Wildlife, for assistance with grant administration. This study is a contribution from the Ohio Biodiversity Conservation Partnership.

Additional Information and Declarations

Competing Interests

Author Contributions

Data Availability

David M. Nelson is an Academic Editor for PeerJ. C.J. Campbell is employed by Bat Conservation International.

Jamin G. Wieringa conceived and designed the experiments, performed the experiments, analyzed the data, prepared figures and/or tables, authored or reviewed drafts of the article, and approved the final draft.

Juliet Nagel conceived and designed the experiments, performed the experiments, prepared figures and/or tables, authored or reviewed drafts of the article, and approved the final draft.

C.J. Campbell conceived and designed the experiments, performed the experiments, analyzed the data, prepared figures and/or tables, authored or reviewed drafts of the article, and approved the final draft.

David M. Nelson conceived and designed the experiments, performed the experiments, prepared figures and/or tables, authored or reviewed drafts of the article, and approved the final draft.

Bryan C. Carstens conceived and designed the experiments, performed the experiments, prepared figures and/or tables, authored or reviewed drafts of the article, and approved the final draft.

H. Lisle Gibbs conceived and designed the experiments, performed the experiments, prepared figures and/or tables, authored or reviewed drafts of the article, and approved the final draft.

The following information was supplied regarding data availability:

All raw trace data is available at Dryad: Wieringa, Jamin (2024). Data from: Geographic source of bats killed at wind-energy facilities in the eastern United States [Dataset]. Dryad. https://doi.org/10.5061/dryad.kd51c5bcr.

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
