# Peer review of "Geographic source of bats killed at wind-energy facilities in the eastern United States"

_PeerJ, doi:10.7717/peerj.16796_

## Round 0.1 · original submission · Minor Revisions

Dear authors,

Following a thorough review by two esteemed reviewers, it is gratifying to note that both concur on the manuscript's intriguing nature and its merit for publication. Nevertheless, minor revisions are deemed essential to ensure its acceptance.

Best regards,

Armando Sunny

·

Basic reporting

The manuscript is well written with a short introduction to the problem and the relevant natural history characteristics of these bats.

Experimental design

The study design is appropriate for the questions addressed in this manuscript.

Validity of the findings

The findings appear to be valid, based on the description of the methods and results.

Additional comments

This manuscript describes an analysis of combining stable isotopes, trace elements, and species distribution models to assess the geographic origins of migratory bats. The manuscript is well-written and the presentation and approach is logical. I don’t see anything of note that needs to be revised related to this paper and will suggest acceptance. I didn’t spend a lot of time with the paper, as the species distribution modeling details were addressed in a prior paper by this group that I reviewed. The stable isotope and trace element analyses are outside of my areas of expertise.

Reviewer 2 ·

Basic reporting

All my comments are submitted together in 'Additional Comments.'

Experimental design

All my comments are submitted together in 'Additional Comments.'

Validity of the findings

All my comments are submitted together in 'Additional Comments.'

Additional comments

This is a well-written paper that gets to a really interesting question in bat ecology: to what degree do wind-related bat fatalities affect the local vs. migratory population? This is a well-considered study that builds on an updated approach for location assignment in another recent paper. While this study is similar to the methods paper, the results and discussion are sufficiently focused on the biological patterns that it is different enough to have independent scientific value.

I have a few high-level concerns that probably are being addressed to the best degree possible, but it’s worth discussing them further in the paper. First, is the use of occupancy SDMs to aid in migratory origin assignment. As mentioned, I’m not sure how there is a better option available for bats, but the use of occupancy models could bias the origin estimates. If occupancy probability is strongly correlated with density, then the issues will be minimal but if it does not then you are likely undervaluing the number of animals coming from locations with high occupancy probability. It would be helpful to add a discussion of this sort of bias.

Framing the paper as an assessment of residency seemed counterintuitive to me. Residency testing is always going to be inferentially fraught as you are testing for no difference. As you mention in the paper, there are loads of reasons for no difference to be found between any two samples and this leads to a lot of hedging in the paper. While that hedging is certainly needed, it seems earlier to frame the paper in terms of the migrants. Those individuals are confirmed to have arrived from non-local summering areas and centering the inference on the successful determination of non-local origin seems more straightforward to me.

Finally, I was reviewing the Wieringa et al. (2023) Ecosphere paper and the trace element plots in some of the example plots (e.g., Fig. 4) looked absolutely wild to me. It’s clear that adding these data is helpful to the assignment modeling, but seeing those extremely disjointed prediction surfaces made me think that something very strange was happening in those spatial models. If you think those artifacts are realistic to the environmental distributions of those elements, then maybe this isn’t worth commenting on. But those models made me think something strange was going on and it might be worth mentioning as a source of error.

Nice job with the paper, I thought it was an interesting read. The quantitative approaches seemed sound (though I didn’t review the Ecosphere paper completely) and the biological inference was well-founded.



Line-by-line comments:

18. This seems like an understatement given Frick et al. (2017), though origins would certainly influence her analysis in that paper.

22. The ‘most likely summer geographic origins…’

26. The big trick to this appears to be what you want to call ‘residents’. In reality, they are individuals where you lack evidence of movement, but what scale do you actually have the data to assess movements? It seems likely that some of these individuals are migrants, but from within the range of locations that you cannot differentiate with your data sources.

36. I would say ‘mitigate’ climate change as I don’t think war metaphors are very helpful, but your mileage may vary.

62. Yes, this is the issue with calling them ‘residents.’ You could call it ‘apparent migratory state’ to reduce some confusion here. Also, I would be clear about what ‘a long distance’ is and how that might vary across species.

142. I got a little confused in this paragraph. Are you describing how this R function works later on, or another process entirely? Generally, I think it’s better to describe the method than reference the R function that helped you do it. In this case, clearly explaining how the method determines if the animal is from the local area would be helpful as it’s the crux of your inference.

155. I would just take the maximum value here and be (probably) overly cautious in your assignments.

164-165. I think I know what you mean here, but the 95th percentile of what exactly? The second sentence (starting ‘Using the average…) is presented with no evidence. It certainly distributes the likely origin across multiple locations, but it’s unclear what ‘a better estimate’ means and you are combining cells of a variety of likelihoods. It’s not always going to be straightforwardly better. I’m curious if this actually solves the problem that is stated earlier in the paragraph. If there is a suite of disparate locations that the model thinks are good, is taking the average of them actually helpful? Does that select a location that could have a much lower likelihood than any of the preferred cells?

209. Did you test the differences in resident mortality across sites? I agree with the trend, but you didn’t formally test the effect from what I can see.

213. This seems like a good opportunity to define the spatial scope of your uncertainty through their entire modeling process.

285. These differences could be due to the averaging that you are doing across multiple likely locations. I’m not sure your explanations are as salient. SDMs are increasing the accuracy of the predictions, right? So how is including them hurting what is fundamentally an accuracy issue?

306. Agreed. And thinking about how to optimize the conservation impacts of curtailment through modulating cut-in speed in blanket (Adams et al. 2021) or more complex algorithm driven approaches (Barré et al. 2023). Thinking about how that interacts with these findings about migratory status to maximize conservation impact would be interesting.

Table 2. You misspelled ‘percentile’

Figures 1 and 2. I’m not a huge fan of pie charts, they tend to misrepresent data. I would use bar charts instead.

Literature cited

Adams, E.M., Gulka, J. and Williams, K.A., 2021. A review of the effectiveness of operational curtailment for reducing bat fatalities at terrestrial wind farms in North America. PLoS One, 16(11), p.e0256382.

Barré, K., Froidevaux, J.S., Sotillo, A., Roemer, C. and Kerbiriou, C., 2023. Drivers of bat activity at wind turbines advocate for mitigating bat exposure using multicriteria algorithm-based curtailment. Science of the Total Environment, 866, p.161404.

---

## Round 0.2 · accepted · Accept

Dear Authors,

I am pleased to inform you that both reviewers have concurred on the appropriateness of the corrections implemented in your manuscript. As a result, it is with great satisfaction that I announce the acceptance of your work for publication in PeerJ. I extend my heartfelt gratitude for the meticulous attention to detail demonstrated in your revisions.

Thank you for selecting PeerJ as the platform for sharing your compelling and valuable research. Your contribution is integral to advancing our understanding and promoting the conservation of this species.

Warm regards,

Armando Sunny

Reviewer 2 ·

Basic reporting

All revisions look reasonable.

Experimental design

All revisions look reasonable.

Validity of the findings

All revisions look reasonable.

Additional comments

All revisions look reasonable.